# Genome-wide association and expression quantitative trait loci studies identify multiple susceptibility loci for thyroid cancer

Ho-Young Son[1,*], Yul Hwangbo[2,3,*], Seong-Keun Yoo[4,5], Sun-Wha Im[1], San Duk Yang[4], Soo-Jung Kwak[4], Min Seon Park[1,6], Soo Heon Kwak[2], Sun Wook Cho[2], Jun Sun Ryu[3], Jeongseon Kim[7], Yuh-Seog Jung[3], Tae Hyun Kim[3], Su-jin Kim[8], Kyu Eun Lee[8], Do Joon Park[2], Nam Han Cho[9], Joohon Sung[10], Jeong-Sun Seo[1,4,5], Eun Kyung Lee[3,**], Young Joo Park[2,5,**] & Jong-Il Kim[1,4,11,**]

Thyroid cancer is the most common cancer in Korea. Several susceptibility loci of differentiated thyroid cancer (DTC) were identified by previous genome-wide association studies (GWASs) in Europeans only. Here we conducted a GWAS and a replication study in Koreans using a total of 1,085 DTC cases and 8,884 controls, and validated these results using expression quantitative trait loci (eQTL) analysis and clinical phenotypes. The most robust associations were observed in the *NRG1* gene (rs6996585, $P = 1.08 \times 10^{-10}$) and this SNP was also associated with *NRG1* expression in thyroid tissues. In addition, we confirmed three previously reported loci (*FOXE1*, *NKX2-1* and *DIRC3*) and identified seven novel susceptibility loci (*VAV3*, *PCNXL2*, *INSR*, *MRSB3*, *FHIT*, *SEPT11* and *SLC24A6*) associated with DTC. Furthermore, we identified specific variants of DTC that have different effects according to cancer type or ethnicity. Our findings provide deeper insight into the genetic contribution to thyroid cancer in different populations.

[1] Department of Biochemistry and Molecular Biology, Seoul National University College of Medicine, 103 Daehak-ro, Jongno-gu, Seoul 03080, Republic of Korea. [2] Department of Internal Medicine, Seoul National University College of Medicine, Seoul 03080, Republic of Korea. [3] Center for Thyroid Cancer, National Cancer Center, Goyang 10408, Republic of Korea. [4] Department of Biomedical Sciences, Seoul National University Graduate School, Seoul 03080, Republic of Korea. [5] Genomic Medicine Institute, Medical Research Center, Seoul National University, Seoul 03080, Republic of Korea. [6] Graduate Program in Genetic Counseling, Northwestern University, Chicago, Illinois 60637, USA. [7] Molecular Epidemiology Branch, Division of Cancer Epidemiology and Prevention, Research Institute, National Cancer Center, Goyang 10408, Republic of Korea. [8] Department of Surgery, Seoul National University College of Medicine, Seoul 03080, Republic of Korea. [9] Department of Preventive Medicine Ajou University School of Medicine, Suwon 16499, Republic of Korea. [10] Department of Epidemiology and Institute of Environment and Health, School of Public Health, Seoul National University, Seoul 08826, Republic of Korea. [11] Cancer Research Institute, Seoul National University College of Medicine, Seoul 03080, Republic of Korea. * These authors equally contributed to this work. ** These authors jointly supervised this work. Correspondence and requests for materials should be addressed to E.K.L. (email: waterfol@ncc.re.kr) or to Y.J.P. (email: yjparkmd@snu.ac.kr) or to J.-I.K. (email: jongil@snu.ac.kr).

Thyroid cancer represents the most commonly diagnosed cancer in Korea[1], and differentiated thyroid cancer (DTC) comprises ~90% of thyroid cancers[2]. Thyroid cancers have a high degree of heritability and genetic effects have been reported to account for 53% of the causation of thyroid cancer[3]. The prevalence of familial DTC accounts for ~4–5% of patients with thyroid cancers of follicular cell origin in western countries[4,5]. In Korea, the prevalence of familial DTC was also high and accounts for 9.6% (ref. 6) of cases, suggesting that the genetic susceptibility to DTC differs from that in western countries.

Recently, the advancements in genomic technology have facilitated large-scale genetic studies for diverse diseases. The genome-wide association study (GWAS) has emerged as a popular method to identify genetic factors involved in complex diseases, including several types of cancers[7]. In thyroid cancer, several GWASs were conducted in European descent patients[8–14] and some single-nucleotide polymorphisms (SNPs) associated with thyroid cancer risk were identified, including markers near FOXE1, NKX2-1, DIRC3, NRG1, IMMP2L, RARRES1, SNAPC4, BATF, DHX35, GALNTL4, HTR1B and FOXA2. Among these, only the signals at FOXE1, NKX2-1, DIRC3 and NRG1 were confirmed in different populations, including Japanese, Chinese, Polish and British, by targeted genotyping methods[15–18]. Moreover, even in previous GWAS, there seems to be population differences of the associated SNPs for thyroid cancer. Figlioli et al.[12] reported that two loci discovered by GWAS were replicated only in the Italian population, but not in the Polish and the Spanish populations, suggesting between-study heterogeneity in GWAS of thyroid cancer. All of these findings support the possibility of differences in the genetic factors among different populations.

Among the identified SNPs, the rs965513 polymorphism, which is the most robust among the previously identified locus (FOXE1, 9q22.33), was associated with a significantly increased risk of thyroid cancer with an odds ratio (OR) of 1.80 in Caucasians in a meta-analysis. In contrast, an analysis in East Asians yielded an OR of 1.35 (ref. 19). Furthermore, the frequency of the risk allele of the rs965513 polymorphism in a general population was widely varied across different ethnicities, ranging from 0.06 to 0.44. The frequency in East Asians was <0.1, which is much lower than 0.39 in Caucasians[19]. This suggests that the genetic contribution of previously identified susceptibility loci to development of thyroid cancer might be small in East Asian, even though GWAS for thyroid cancer have not been conducted in Asian populations yet. In addition, an expression quantitative trait loci (eQTL) analysis using RNA-sequencing data from thyroid tissue for the identified loci has not been conducted in patients with thyroid cancer. Therefore, we conducted both GWAS and eQTL studies to identify potential susceptibility loci for thyroid cancer in Koreans and identified a strong association of the NRG1 gene and a modest association of six new genes in this population.

## Results

**Stage 1 genome scan.** After genotype imputation, quality control and the removal of the batch effect and relatedness (see Methods), we conducted an association analysis using 3,593,389 markers for DTC, papillary thyroid cancer (PTC) and follicular thyroid cancer (FTC) versus the control (Supplementary Fig. 1 and Supplementary Table 1). A quantile–quantile (Q–Q) plot and genomic inflation factors showed little evidence for statistic inflation (Supplementary Fig. 2). The genome-wide association results of each of the DTC and PTC cases are shown in the Manhattan plots (Fig. 1). We identified two genome-wide significant ((additive model of logistic regression) $P = 5 \times 10^{-8}$)

loci in DTC and two loci in PTC. In the test using the DTC cases, the most significantly associated SNP was observed near the FOXE1 gene (rs965513, (additive model of logistic regression) $P = 2.35 \times 10^{-9}$) at 9q22.33. The second significant SNP was located in the 3′-untranslated region of the SLC24A6 gene (rs16934253, (additive model of logistic regression) $P = 2.49 \times 10^{-9}$) at 12q24.13 (Fig. 1a). In the test using the PTC cases, we found two significantly associated signals in the intronic region of the NRG1 gene (rs6996585, (additive model of logistic regression) $P = 5.17 \times 10^{-10}$) at 8p12 and near the FOXE1 gene (rs965513, (additive model of logistic regression) $P = 8.20 \times 10^{-9}$), which was the most significant SNP in DTC (Fig. 1b). Forty-one candidate SNPs were selected for the replication (Supplementary Table 3). Among these, only six SNPs of DIRC3, NKX2-1 or NRG1 were identical to the previous reports (Supplementary Table 2).

**Stage 2 follow-up and joint Stages 1 and 2 analyses.** Among the 41 candidate SNPs, 13 SNPs in 10 loci were replicated in DTC or PTC and 2 SNPs in 2 loci were replicated in FTC (green coloured loci in Fig. 1). The minor allele frequency (MAF) of all of the 15 SNPs in our control samples were similar with those of the East Asian population in 1,000 genomes (Supplementary Table 4), with the exception of rs1549738. Table 1 describes the replicated SNPs in DTC, PTC and FTC through the GWAS, and the replication and joint association analysis. Most of the SNPs showed a similar association between DTC and PTC, with the exception of 2 SNPs; rs4915076 of VAV3 and rs9858271 of FHIT were replicated only in PTC. After the joint analysis, the most significantly associated region was the NRG1 loci and the second one was DIRC3.

Regarding the candidate SNPs that were associated with FTC, despite the limited number of FTC samples (discovery $N = 60$, replication $N = 28$), we identified two SNPs that were highly associated with FTC but were not positively replicated in the DTC or PTC samples. The SNP (rs16934253) in the 3′-untranslated region of SLC24A6 at 12q24.13 is the second most significantly associated signal of DTC in the discovery stage, but it was not well replicated in DTC or PTC. However, we identified that rs16934253 showed a high-risk effect ((additive model of logistic regression) Joint $P = 2.71 \times 10^{-5}$, OR = 3.32) in the FTC samples. Another candidate SNP (rs1549738) near DIRC3 showed a similar risk effect in the FTC samples ((additive model of logistic regression) Joint $P = 0.0017$, OR = 1.65) but not in the DTC or PTC samples.

Among the SNPs, two SNPs in two loci were identical (rs2439302 in the NRG1 locus and rs944289 in the NKX2-1 locus) and six SNPs in four loci (rs6996585 and rs12542743 in the NRG1 locus, rs12990503 and rs1549738 in the DIRC3 locus, rs34081947 in the NKX2-1 locus and rs72753537 in the FOXE1 locus) were located at the same loci that were identified in previous reports (Supplementary Table 1). The other seven SNPs in seven loci (VAV3, PCNXL2, INSR, MRSB3, FHIT, SEPT11 and SLC24A6 loci) were newly identified.

**Validation of the candidate SNPs with cis-eQTL and GSEA.** Next, we conducted a cis-eQTL analysis with the 15 positively replicated SNPs and their nearby genes using 78 tumour and 23 normal thyroid tissues from samples of the replication stage (Table 2). We found significant cis-eQTL of genes near NRG1, NKX2-1, DIRC3, PCNXL2 and VAV3 in the normal and tumour thyroid tissue. We also evaluated the cis-eQTL of the normal thyroid tissues (Supplementary Table 5) and various other tissues (Supplementary Table 6) in the public eQTL database and imputed the expression (Table 2) of 470 DTC case samples using

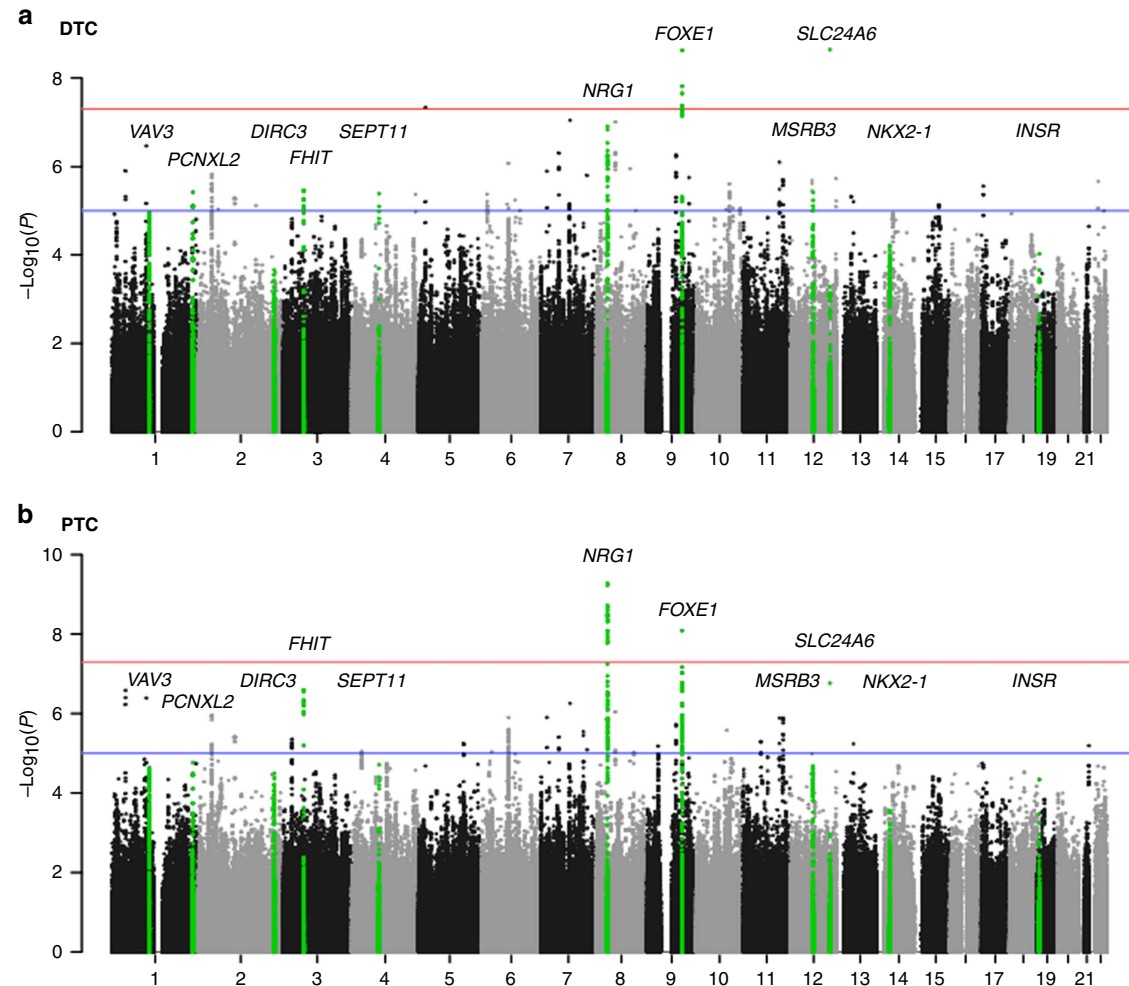

**Figure 1 | Genome-wide association plot for thyroid cancer.** The Manhattan plots of genome-wide association signals with (**a**) DTC and (**b**) PTC for stage 1. The x-axis represents the SNP markers on each chromosome. The y-axis shows the $-\log_{10}$ P-value (logistic regression). The red horizontal line represents the genome-wide significance threshold $P = 5.0 \times 10^{-8}$, and the blue horizontal line represents the genome-wide suggestiveness threshold $P = 1.0 \times 10^{-6}$. Eleven candidate loci of DTC, PTC or FTC are shown in green.

PrediXcan, found similar results to our RNAseq eQTL data. To reveal the transcriptional change and biological function during the cancer prognosis, we conducted a gene set enrichment analysis (GSEA) according to the candidate SNPs in the tumour and normal thyroid tissue. We found several cellular growths or cancer-related pathways that were associated with SNPs of *NRG1*, *VAV3*, *DIRC3*, *SEPT11* and *INSR* (Supplementary Table 7). Specifically, rs6996585 of *NRG1* was observed in association with a number of those pathways in the normal thyroid tissue (Supplementary Table 8).

**Association between candidate SNPs and clinical phenotypes.** We further investigated whether there was an association between candidate SNPs and clinical phenotypes, such as the *BRAF*^V600E mutation, lymph node metastasis or extrathyroidal extension (Supplementary Table 9). Interestingly, three SNPs of *NRG1* were associated with lymph node metastasis in the *BRAF*^V600E positive samples. The SNPs of *SEPT11* and *INSR* were associated with extrathyroidal extension.

**The most significantly associated variant in the NRG1 locus.** The most significant association was identified in the intronic region of *NRG1* at 8p12. In this locus, SNPs on the *NRG1* gene were shown to have a more significant association with PTC than

with DTC (Figs 1 and 2, and Table 1). In a joint analysis, rs6996585 of *NRG1* was the most significant signal ((additive model of logistic regression) $P = 9.01 \times 10^{-12}$ in PTC, $1.08 \times 10^{-10}$ in DTC). We found a significant *cis*-eQTL of rs6996585 for *NRG1* expression in thyroid tumour tissue ((additive model of linear regression) $P = 0.0053$, Fig. 2c and Table 2). A similar expression pattern was observed in normal thyroid tissue, although it was not statistically significant ((additive model of linear regression) $P$-value = 0.0526). However, the predicted expression results for the discovery case samples that used the normal thyroid reference data showed a highly significant association ((additive model of linear regression) $P = 2.99 \times 10^{-244}$, Fig. 2c and Table 2). The public expression data also showed a significant association ((additive model of linear regression) $P = 5.79 \times 10^{-21}$) in the normal thyroid tissue (Supplementary Fig. 3a and Supplementary Table 5). Furthermore, the GSEA result indicated that rs6996585 was significantly associated with 31 gene set pathways related to cellular growth signals or cancer in the normal thyroid tissue (false discovery rate $q < 0.05$, Supplementary Tables 7 and 8). We confirmed that the common genes from 31 significant gene sets were enriched in the *ERBB-MAPK* signalling pathway (Fig. 2f). A clinical phenotype analysis showed that rs6996585 was associated with lymph node metastasis in patients

**Table 1 | DTC-, PTC- and FTC-associated SNPs in Korean population.**

| SNP | Chr Position Gene | Risk allele | Stage | DTC Allele frequency (case/control) | OR | P-value | PTC Allele frequency (case/control) | OR | P-value | FTC Allele frequency (case/control) | OR | P-value |
|---|---|---|---|---|---|---|---|---|---|---|---|---|
| rs6996585 | 8 32400803 NRG1 | G | Discovery Replication Joint | 0.30/0.23 0.28/0.23 0.29/0.23 | 1.48 1.29 1.39 | 1.20E − 07 0.0061 1.08E − 10 | 0.32/0.23 0.28/0.23 0.29/0.23 | 1.61 1.28 1.43 | 5.17E − 10 0.0094 9.01E − 12 | 0.17/0.23 0.32/0.23 0.22/0.23 | 0.70 1.59 0.96 | 0.1499 0.1130 0.8273 |
| rs12542743 | 8 32318355 NRG1 | C | Discovery Replication Joint | 0.32/0.25 0.31/0.27 0.32/0.25 | 1.42 1.22 1.36 | 1.12E − 06 0.0267 4.61E − 10 | 0.34/0.25 0.31/0.27 0.32/0.25 | 1.53 1.20 1.39 | 1.63E − 08 0.0427 1.01E − 10 | 0.21/0.25 0.38/0.27 0.26/0.25 | 0.78 1.63 1.04 | 0.2674 0.0821 0.8137 |
| rs2439302 | 8 32432369 NRG1 | G | Discovery Replication Joint | 0.27/0.21 0.27/0.21 0.27/0.21 | 1.36 1.37 1.37 | 8.38E − 05 8.55E − 04 1.42E − 09 | 0.29/0.21 0.27/0.21 0.28/0.21 | 1.48 1.36 1.41 | 1.50E − 06 0.0013 1.26E − 10 | 0.15/0.21 0.30/0.21 0.20/0.21 | 0.66 1.59 0.94 | 0.1129 0.1143 0.7289 |
| rs12990503 | 2 218294217 DIRC3 | G | Discovery Replication Joint | 0.68/0.62 0.70/0.65 0.69/0.63 | 1.32 1.21 1.34 | 1.82E − 04 0.0268 3.55E − 09 | 0.70/0.62 0.70/0.65 0.70/0.63 | 1.38 1.24 1.38 | 3.17E − 05 0.0164 2.58E − 10 | 0.61/0.62 0.61/0.65 0.61/0.63 | 0.93 0.83 0.93 | 0.7107 0.5119 0.6324 |
| rs11175834 | 12 65992636 MSRB3 | T | Discovery Replication Joint | 0.21/0.15 0.19/0.14 0.20/0.15 | 1.45 1.41 1.37 | 1.16E − 05 0.0018 4.26E − 08 | 0.21/0.15 0.19/0.14 0.20/0.15 | 1.44 1.38 1.36 | 4.52E − 05 0.0035 4.86E − 07 | 0.21/0.15 0.25/0.14 0.22/0.15 | 1.48 1.99 1.60 | 0.0879 0.0289 0.0100 |
| rs4915076 | 1 108359505 VAV3 | T | Discovery Replication Joint | 0.77/0.70 0.75/0.71 0.76/0.70 | 1.42 1.20 1.33 | 9.37E − 06 0.0507 8.47E − 08 | 0.77/0.70 0.75/0.71 0.76/0.70 | 1.41 1.23 1.34 | 4.51E − 05 0.0240 7.09E − 08 | 0.77/0.70 0.63/0.71 0.73/0.70 | 1.48 0.68 1.14 | 0.0707 0.1637 0.4311 |
| rs4649295 | 1 233416538 PCNXL2 | T | Discovery Replication Joint | 0.88/0.82 0.86/0.82 0.87/0.82 | 1.56 1.33 1.43 | 1.04E − 05 0.0106 6.00E − 08 | 0.88/0.82 0.86/0.82 0.87/0.82 | 1.56 1.36 1.45 | 3.48E − 05 0.0068 8.53E − 08 | 0.87/0.82 0.80/0.82 0.85/0.82 | 1.54 0.89 1.27 | 0.1155 0.7326 0.2634 |
| rs34081947 | 14 36559531 NKX2-1 | T | Discovery Replication Joint | 0.47/0.41 0.47/0.39 0.47/0.41 | 1.28 1.38 1.27 | 2.40E − 04 8.07E − 05 1.19E − 07 | 0.47/0.41 0.47/0.39 0.47/0.41 | 1.24 1.37 1.25 | 0.003163 1.31E − 04 2.47E − 06 | 0.53/0.41 0.50/0.39 0.52/0.41 | 1.62 1.56 1.56 | 0.0079 0.0999 0.0030 |
| rs1874564 | 4 77858105 SEPT11 | G | Discovery Replication Joint | 0.77/0.69 0.74/0.69 0.75/0.69 | 1.44 1.24 1.31 | 3.43E − 06 0.0169 2.04E − 07 | 0.77/0.69 0.74/0.69 0.75/0.69 | 1.43 1.25 1.31 | 1.93E − 05 0.0135 5.87E − 07 | 0.77/0.69 0.69/0.69 0.75/0.69 | 1.51 1.00 1.32 | 0.0576 0.9930 0.1171 |
| rs9858271 | 3 59545330 FHIT | G | Discovery Replication Joint | 0.50/0.43 0.47/0.43 0.48/0.43 | 1.37 1.15 1.26 | 3.57E − 06 0.0952 6.82E − 07 | 0.52/0.43 0.47/0.43 0.49/0.43 | 1.45 1.18 1.30 | 2.78E − 07 0.0468 2.76E − 08 | 0.41/0.43 0.32/0.43 0.38/0.43 | 0.92 0.62 0.82 | 0.6570 0.0989 0.2029 |
| rs944289 | 14 36649246 NKX2-1 | T | Discovery Replication Joint | 0.51/0.46 0.51/0.43 0.51/0.46 | 1.24 1.38 1.25 | 0.0014 7.53E − 05 1.39E − 06 | 0.51/0.46 0.51/0.43 0.51/0.46 | 1.22 1.36 1.23 | 0.0062 1.93E − 04 1.72E − 05 | 0.54/0.46 0.59/0.43 0.56/0.46 | 1.40 1.90 1.50 | 0.0646 0.0186 0.0072 |
| rs72753537 | 9 100660746 FOXE1 | C | Discovery Replication Joint | 0.12/0.07 0.09/0.07 0.10/0.07 | 1.63 1.38 1.41 | 3.56E − 06 0.0352 7.67E − 06 | 0.12/0.07 0.09/0.07 0.11/0.07 | 1.76 1.43 1.48 | 1.70E − 07 0.0209 5.37E − 07 | 0.06/0.07 0.04/0.07 0.05/0.07 | 0.77 0.52 0.67 | 0.4958 0.3560 0.2448 |
| rs7248104 | 19 7224431 INSR | A | Discovery Replication Joint | 0.43/0.36 0.40/0.35 0.41/0.36 | 1.31 1.20 1.22 | 6.77E − 05 0.0313 2.00E − 05 | 0.43/0.36 0.40/0.35 0.41/0.36 | 1.35 1.20 1.23 | 4.57E − 05 0.0293 1.64E − 05 | 0.39/0.36 0.38/0.35 0.38/0.36 | 1.11 1.10 1.09 | 0.5877 0.7449 0.5731 |
| **FTC-associated SNPs** | | | | **DTC** | | | **PTC** | | | **FTC** | | |
| rs16934253 | 12 113737225 SLC24A6 | A | Discovery Replication Joint | 0.05/0.02 0.02/0.02 0.03/0.02 | 2.46 0.98 1.51 | 2.49E − 09 0.9573 0.0016 | 0.05/0.02 0.02/0.02 0.03/0.02 | 2.36 0.83 1.36 | 1.71E − 07 0.5752 0.0216 | 0.07/0.02 0.07/0.02 0.07/0.02 | 3.20 4.35 3.32 | 8.95E − 04 0.0045 2.71E − 05 |
| rs1549738 | 2 218118722 DIRC3 | A | Discovery Replication Joint | 0.61/0.55 0.56/0.56 0.58/0.55 | 1.28 1.04 1.14 | 2.96E − 04 0.6595 0.0036 | 0.61/0.55 0.56/0.56 0.58/0.55 | 1.25 1.01 1.11 | 0.0026 0.9003 0.0307 | 0.66/0.55 0.70/0.56 0.67/0.55 | 1.56 1.84 1.65 | 0.0193 0.0376 0.0017 |

Chr, chromosome number; DTC, differentiated thyroid cancer; FTC, follicular thyroid cancer; OR, odds ratio; PTC, papillary thyroid cancer; SNP, single-nucleotide polymorphism.
The SNP positions are indexed to the National Center for Biotechnology Information (NCBI) build 37. Association results generated from additive model of logistic regression analyses.

with $BRAF^{V600E}$ mutated tumours ((additive model of linear regression) $P = 0.015$, Fig. 2g and Supplementary Table 9).

In this region, another candidate SNP (rs12542743) had a similar association ((additive model of logistic regression) Joint $P = 1.01 \times 10^{-10}$ in PTC, $4.61 \times 10^{-10}$ in DTC, Fig. 2a,b) and cis-eQTL results (Fig. 2d and Table 2). Although the previously reported SNP (rs2439302) showed a marginal association ((additive model of logistic regression) $P = 1.50 \times 10^{-6}$ in PTC, $8.38 \times 10^{-5}$ in DTC) in the discovery stage, the cis-eQTL of this SNP was more significantly associated than that of rs6996585 (Fig. 2e and Supplementary Fig. 3c). These two SNPs also showed a similar association with lymph node metastasis in the $BRAF^{V600E}$ mutation-positive group (Supplementary Table 9).

**Table 2 | *Cis*-eQTL results of candidate SNPs and nearby genes.**

| Chr | SNP | Position | Representative gene | RNA-sequencing data in this study | | | Predicted expression of 470 DTC cases using normal thyroid reference | |
|---|---|---|---|---|---|---|---|---|
| | | | | *Cis*-eQTL gene | *P*-value of tumour tissue | *P*-value of normal tissue | *Cis*-eQTL gene | *P*-value |
| 1 | rs4915076 | 108359505 | *VAV3* | *VAV3* | **0.0174** | 0.0995 | *VAV3* | **< 1.00E − 300** |
| 1 | rs4649295 | 233416538 | *PCNXL2* | *PCNXL2* | **0.0030** | 0.8594 | *PCNXL2* | **3.81E − 05** |
| | | | | *NTPCR* | 0.9006 | **0.0472** | *NTPCR* | **0.0262** |
| 2 | rs1549738 | 218118722 | *DIRC3* | *TNS1* | **0.0023** | 0.1170 | *TNS1* | NA |
| 2 | rs12990503 | 218294217 | *DIRC3* | − | | | − | |
| 3 | rs9858271 | 59545330 | *FHIT* | − | | | − | |
| 4 | rs1874564 | 77858105 | *SEPT11* | − | | | − | |
| 8 | rs6996585 | 32400803 | *NRG1* | *NRG1* | **0.0053** | 0.0526 | *NRG1* | **2.99E − 244** |
| 8 | rs12542743 | 32318355 | *NRG1* | *NRG1* | **0.0073** | 0.1021 | *NRG1* | **3.04E − 97** |
| 8 | rs2439302 | 32432369 | *NRG1* | *NRG1* | **0.0025** | **0.0125** | *NRG1* | **< 1.00E − 300** |
| 9 | rs72753537 | 100660746 | *FOXE1* | *C9orf156* | 0.6914 | 0.3035 | *C9orf156* | **1.09E − 43** |
| | | | | *CORO2A* | 0.0551 | 0.4374 | *CORO2A* | **1.03E − 19** |
| | | | | *XPA* | 0.4269 | 0.9955 | *XPA* | **8.43E − 58** |
| | | | | *TSTD2* | 0.3061 | 0.2419 | *TSTD2* | **0.0013** |
| 12 | rs11175834 | 65992636 | *MSRB3* | − | | | − | |
| 12 | rs16934253 | 113737225 | *SLC24A6* | − | | | − | |
| 14 | rs34081947 | 36559531 | *NKX2-1* | *NKX2-1* | **0.0323** | 0.5458 | *NKX2-1* | NA |
| | | | | *SFTA3* | 0.0883 | 0.4173 | *SFTA3* | **4.05E − 15** |
| 14 | rs944289 | 36649246 | *NKX2-1* | *NKX2-1* | **0.0069** | **0.0302** | *NKX2-1* | NA |
| | | | | *SFTA3* | **0.0107** | **0.0476** | *SFTA3* | **3.90E − 13** |
| | | | | *RALGAPA1* | 0.2493 | 0.2766 | *RALGAPA1* | **0.0172** |
| 19 | rs7248104 | 7224431 | *INSR* | *INSR* | 0.7187 | 0.8680 | *INSR* | **4.91E − 41** |

Chr, chromosome number; DTC, differentiated thyroid cancer; eQTL, expression quantitative trait loci; NA, not available; SNP, single-nucleotide polymorphism.
The SNP positions are indexed to the National Center for Biotechnology Information (NCBI) build 37. The *cis*-eQTL gene is defined as the genes within ±500 kb of the candidate SNP. The *cis*-eQTL results of candidate SNPs are from the association results of 78 tumour thyroid tissues and 23 normal thyroid tissues and estimated via additive model of linear regression analyses. Bold indicates significance of $P < 0.05$.

**Other known associated variants**. At 14q13.3 in a region near *NKX2-1*, two SNPs (rs34081947 and rs944289) were significantly associated with DTC ((additive model of logistic regression) Joint $P = 1.19 \times 10^{-7}$ and $1.39 \times 10^{-6}$, respectively, Supplementary Fig. 4a). The top-ranked variant (rs34081947) in this region showed a significant association ((additive model of linear regression) $P = 0.0323$) with the *NKX2-1* expression level in the tumour tissue (Supplementary Fig. 4d). The rs944289 showed a greater significant association with *NKX2-1* and *SFTA3* expression levels compared with rs34081947 in both the tumour and normal tissues (Supplementary Fig. 4e,f and Table 2).

At 2q35, rs12990503 in the intron of *DIRC3* gene was significantly associated ((additive model of logistic regression) Joint $P = 3.55 \times 10^{-9}$) with DTC (Supplementary Fig. 4b). In the tumour tissue, this SNP showed a *cis*-eQTL for *TNS1* (Supplementary Fig. 4g). We could not find a *cis*-eQTL of *DIRC3* in our data, but the public expression data showed its *cis*-eQTL ((additive model of linear regression) $P = 7.19 \times 10^{-6}$) in skin tissue (Supplementary Fig. 4h and Supplementary Table 6). The GSEA result demonstrated that this SNP was associated with *TGF-β* receptor signalling in the thyroid tumour tissue (Supplementary Table 7).

At 9q22.33 near the *FOXE1* region, rs965513 showed the most significant association in the discovery stage of DTC, but this association was not replicated. Among the seven candidate SNPs previously proposed (Supplementary Table 10), rs72753537 was positively replicated and showed a suggestive association ((additive model of logistic regression) Joint $P = 7.67 \times 10^{-6}$, Supplementary Fig. 4c). This SNP did not have any *cis*-eQTL for nearby genes. However, the predicted expression results showed a *cis*-eQTL with the *C9orf156*, *CORO2A*, *XPA* and *TSTD2* genes (Table 2), and the public expression data showed a *cis*-eQTL with *C9orf156* in various tissues (Supplementary Fig. 4i and Supplementary Table 6). The associations of the seven SNPs in our DTC and PTC subjects are summarized in Supplementary Table 10.

**Novel candidate variants**. Using the expression data, we evaluated the six newly discovered regions. At 1p13.3, an intronic region of *VAV3*, rs4915076, was significantly associated with PTC ((additive model of logistic regression) Joint $P = 7.09 \times 10^{-8}$, Fig. 3a) and is a *cis*-eQTL for *VAV3* expression in tumour tissue ((additive model of linear regression) $P = 0.017$, Fig. 3g). The predicted expression results showed a highly significant association ((additive model of linear regression) $P < 1.00 \times 10^{-300}$, Table 2) and the public expression data also showed a significant association ((additive model of linear regression) $P = 3.33 \times 10^{-27}$) in thyroid tissue (Supplementary Fig. 3d and Supplementary Table 5). The GSEA result showed that rs4915076 was associated with the steroid hormone pathways in the tumour thyroid tissue (Supplementary Table 7). In the intron of *PCNXL2* at

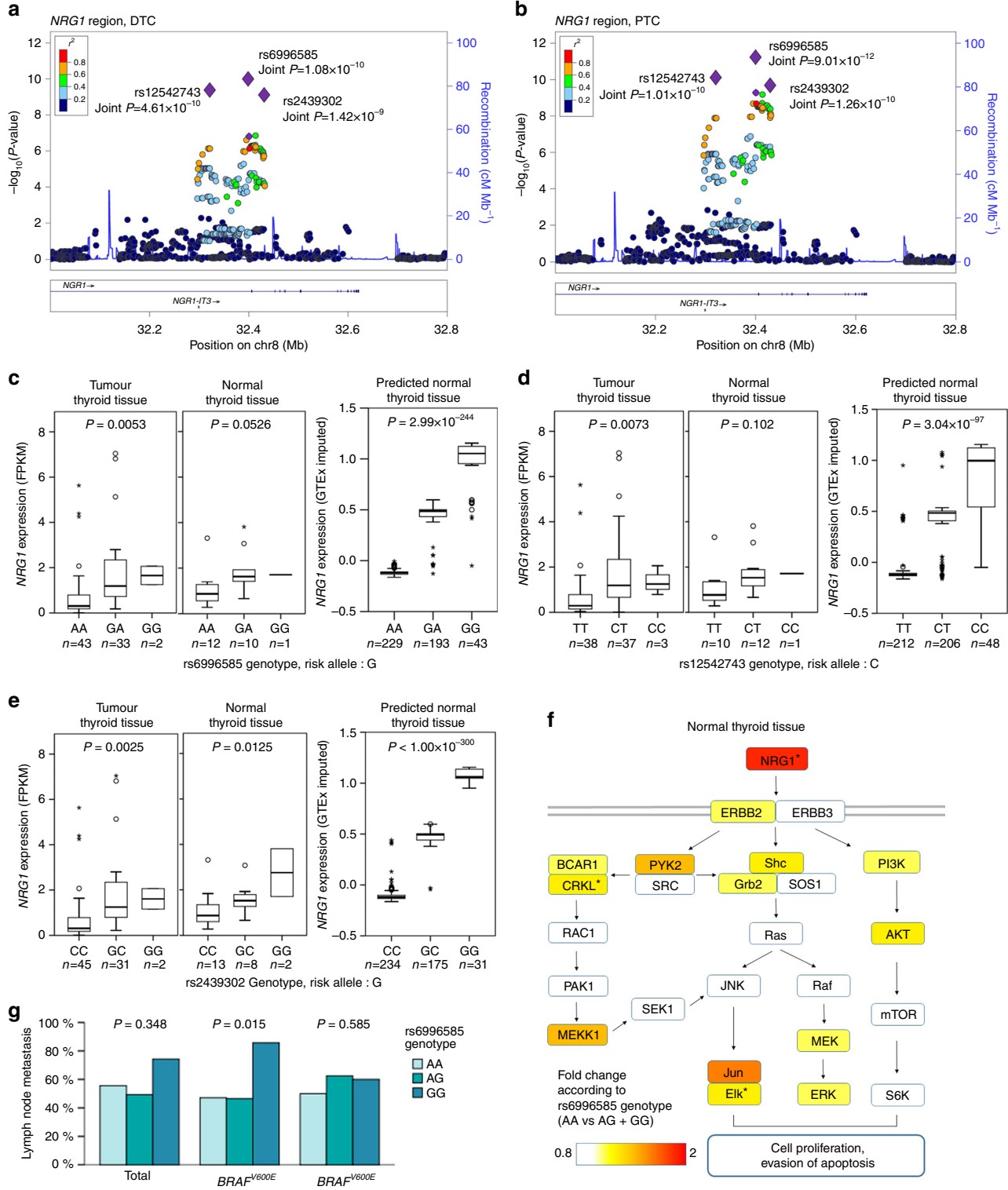

**Figure 2 | Regional association plots and expression of the most associated variant in *NRG1* locus.** Regional association plots for (**a**) DTC and (**b**) PTC. The large purple diamonds indicate the associated SNPs according to joint analyses. Nearby SNPs are colour coded according to the level of LD with the top SNP. The left *y*-axis shows the significance of the association on a − log$_{10}$ *P*-value (logistic regression), and the right *y*-axis shows the recombination rate across the region. Estimated recombination rates from the 1000 Genomes ASN, hg19 database are plotted by the blue line to reflect the local LD structure. The *cis*-eQTL result of *NRG1* in tumour and normal thyroid tissues and the predicted expression of 470 DTC cases according to the (**c**) rs6996585, (**d**) rs12542743 and (**e**) rs2439302 genotypes (additive model of linear regression analyses, error bars represent s.e.m.). (**f**) The characteristic gene expression of normal thyroid tissue. The genes in the *ERBB-MAPK* signalling pathway are represented by the fold change according to the rs6996585 genotype (AA vs AG + GG). Asterisks indicate a significant fold change with a *P*-value <0.05 (*t*-test). (**g**) Lymph node metastasis according to the rs6996585 genotypes and the *BRAF* mutation (additive model of linear regression analyses).

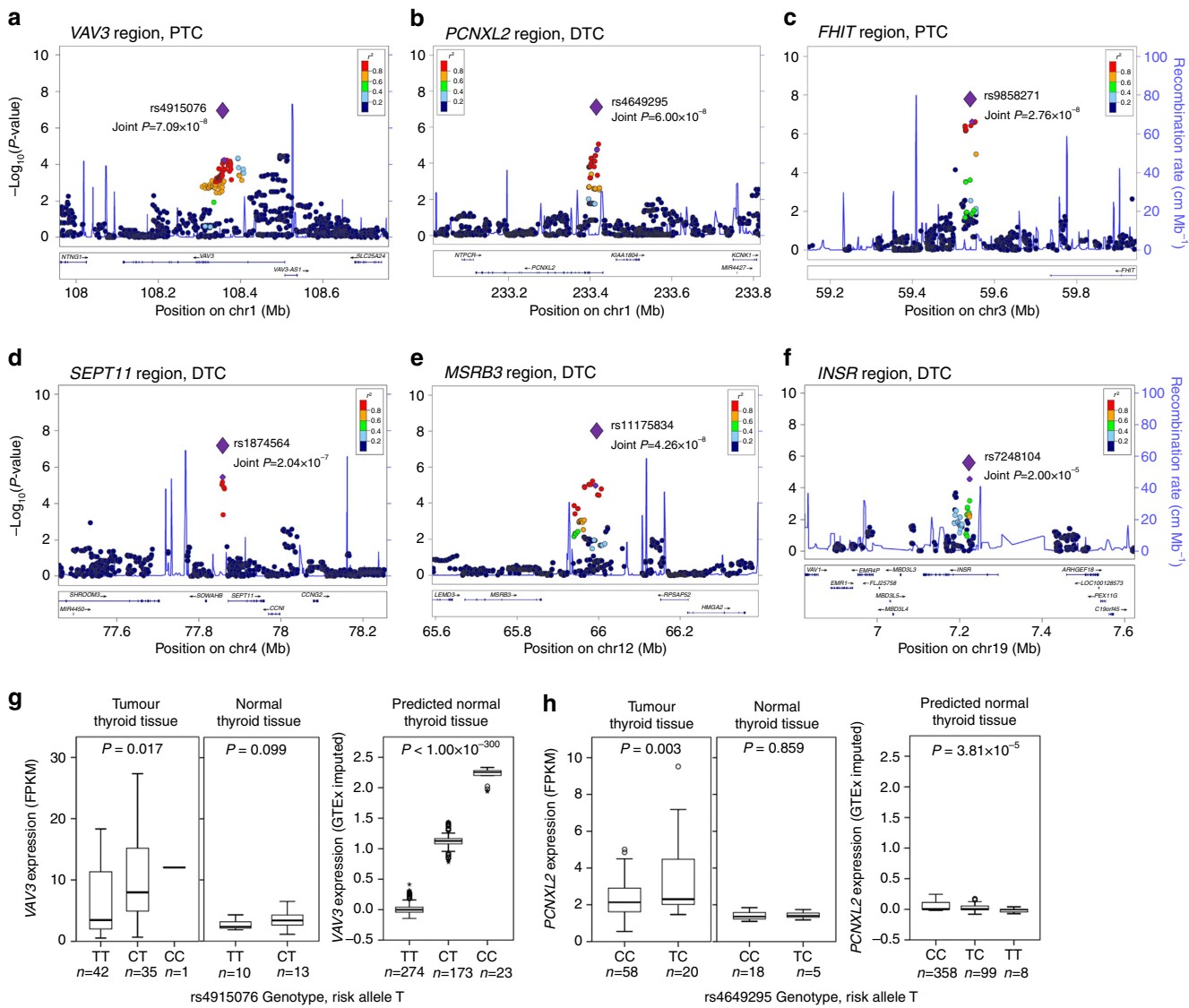

**Figure 3 | Regional association plots and expression of the novel DTC-associated variant.** A regional association plot for the (**a**) *VAV3*, (**b**) *PCNXL2*, (**c**) *FHIT*, (**d**) *SEPT11*, (**e**) *MSRB3* and (**f**) *INSR* regions. The large purple diamonds indicate the most associated SNPs according to joint analyses and nearby SNPs are colour coded according to the level of LD with the top SNP. The left y-axis shows the significance of the association on a $-\log_{10}$ P-value (logistic regression), and the right y-axis shows the recombination rate across the region. Estimated recombination rates from the 1,000 Genomes ASN, hg19 database are plotted by the blue line to reflect the local LD structure. The *cis*-eQTL results of (**g**) *VAV3* and (**h**) *PCNXL2* in tumour and normal thyroid tissues, and the predicted expression of 470 DTC cases according to the rs4915076 and rs4649295 genotypes, respectively (additive model of linear regression analyses, error bars represent s.e.m.).

1q42.2, rs4649295 showed a significant association with DTC ((additive model of logistic regression) Joint $P=6.00 \times 10^{-8}$, Fig. 3b) and is a *cis*-eQTL for *PCNXL2* expression in the tumour tissue ((additive model of linear regression) $P=0.003$, Fig. 3h). Rs9858271, near *FHIT* at 3p14.2, showed a significant association with PTC ((additive model of logistic regression) Joint $P=2.76 \times 10^{-8}$, Fig. 3c). Rs1874564, near *SEPT11* at 4q21.1, showed a significant association with DTC ((additive model of logistic regression) Joint $P=2.04 \times 10^{-7}$, Fig. 3d). On chromosome 12q14.3, an SNP (rs11175834, (additive model of logistic regression) Joint $P=4.26 \times 10^{-8}$) significantly associated with DTC was located near *MSRB3* (Fig. 3e). In the intronic region of the *INSR* gene at 19p13.2, rs7248104 was positively replicated and showed a suggestive association with DTC ((additive model of logistic regression) Joint $P=2.00 \times 10^{-5}$, Fig. 3f). We could not find any *cis*-regulation of nearby gene expression in the *FHIT*, *SEPT11*, *MSRB3* and *INSR* regions based on our expression data;

however, the predicted expression results showed a *cis*-eQTL for *INSR* ((additive model of linear regression) $P=4.91 \times 10^{-41}$, Table 2) and the public expression data showed a *cis*-eQTL for *INSR* in nerve tissue and for *SEPT11* in whole blood (Supplementary Table 6). The GSEA result showed that a SNP of *INSR* was associated with the *ERK* pathway and an SNP of *SEPT11* was associated with the *ATM* pathway (Supplementary Table 7). The association analysis of the clinical phenotypes showed that the SNPs of *SEPT11* and *INSR* were associated with extrathyroidal extension in the $BRAF^{V600E}$ mutation-negative group (Supplementary Table 9).

**A comparison with the European GWAS results.** The effect sizes (OR) and P-values reported in the previous GWAS in Europeans (Supplementary Table 1) were compared with those from our Stage 1 genome scan result (Fig. 4 and Supplementary

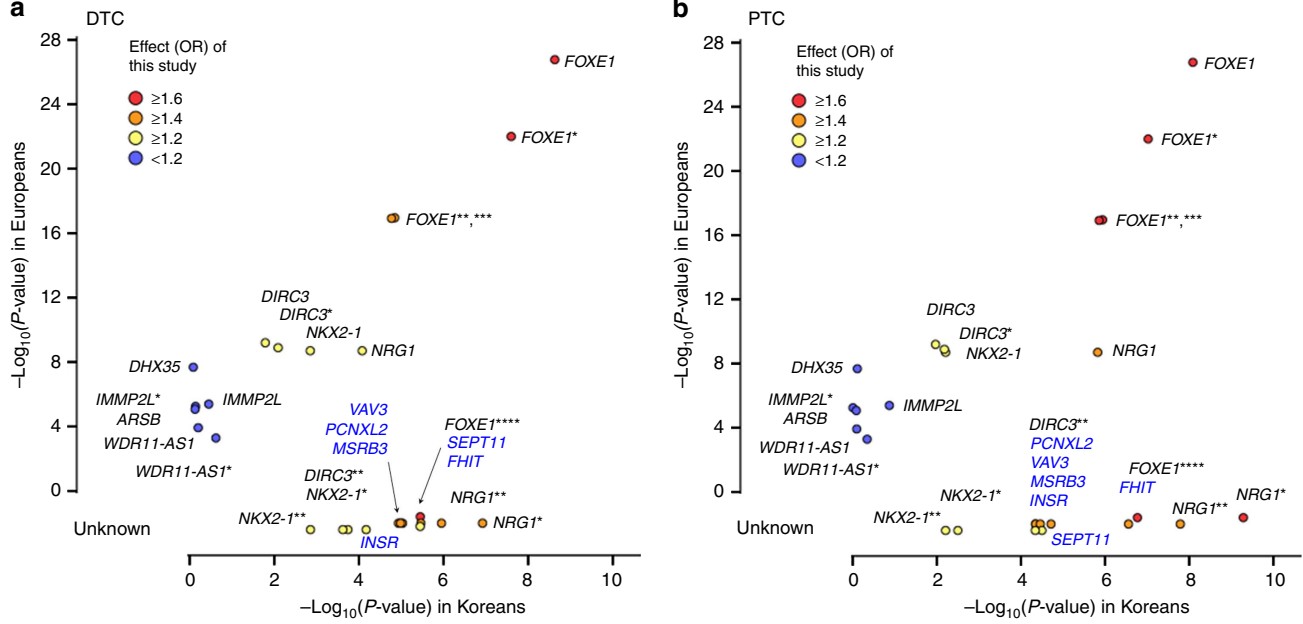

**Figure 4 | Comparison of association result between Europeans and Koreans.** The P-values for (**a**) DTC and (**b**) PTC between Koreans (x-axis) and Europeans (y-axis) are plotted with the corresponding Korean effect sizes (OR; Odd Ratio). The P-value shows the $-\log_{10}$ scale, and the P-values of novel SNPs from this study are compared as unknown. The novel genes of this study are shown in blue. SNPs of the same gene are distinguished by *FOXE1* for rs965513, *FOXE1\** for rs7028661, *FOXE1\*\*\** for rs10122541, *FOXE1\*\*\*\** for rs72753537, *DIRC3* for rs966423, *DIRC3\** for rs6759952, *NRG1* for rs2439302, *NRG1\** for rs6996585, *NRG1\*\** for rs12542743, *NKX2-1* for rs944289, *NKX2-1\** for rs34081947, *NKX2-1\*\** for rs944289, *DIRC3* for rs6759952, *DIRC3\** for rs966423, *DIRC3\*\** for rs12990503, *IMMP2L* for rs10238549, *IMMP2L\** for rs7800391, *WDR11-AS1* for rs2997312 and *WDR11-AS1\** for rs10788123.

Table 11). Fourteen SNPs previously reported in Europeans were available for analyses in our study. Four SNPs of *FOXE1* and one SNP of *NRG1* showed a similar effect size and a similar significant association of $P < 0.0001$. The SNPs of *NKX2-1* and *DIRC3* showed a similar or lesser effect size and a nominal association of $P < 0.05$. The other six SNPs of *IMMP2L*, *DHX35*, *ARSB* and *WDR11-AS1* showed no association in a Korean population.

In addition, the novel associated SNPs of this study were compared with previously reported SNPs. As for DTC, the SNPs of *NRG1* showed a lower association than those of *FOXE1*. However, as for PTC, rs6996585 of *NRG1* showed the most significant P-value of the previous reported SNPs. Furthermore, the novel associated regions (*VAV3*, *PCNXL2*, *INSR*, *MRSB3*, *FHIT* or *SEPT11*) were located in the middle of previously reported SNPs.

## Discussion

In this two-stage GWAS for DTC, we confirmed the associations of the signals at *NRG1*, *NKX2-1*, *FOXE1* and *DIRC3*. Moreover, novel DTC susceptibility loci on *VAV3*, *PCNXL2*, *FHIT*, *SEPT11*, *MSRB3* and *INSR* were identified.

A variant (rs2439302) on *NRG1* was reported be associated with thyroid-stimulating hormone level and DTC in previous GWASs[10,20]. Subsequently, several replication studies validated the association between rs2439302 and thyroid cancer (Supplementary Table 1)[15,17,21]. Neuregulin 1, which is encoded by the *NRG1* gene and acts on the *ERBB* family of tyrosine kinase receptors, could behave as a tumour suppressor in breast cancer cells[22]. One study showed that neuregulin 1 promotes the proliferation and self-renewal of *HER2*-low breast cancers[23]. The intrinsic resistance of PTC to a *BRAF* inhibitor is accompanied by increased *ERBB3* (*HER3*) signalling, which is dependent on *NRG1* autocrine signalling[24]. This *NRG1* dysregulation is closely linked to *PI3K-AKT* and *MAPK* signalling pathway via *ERBB* in lung cancer[25]. Thus, the upregulated *NRG1* expression could be associated with thyroid cancer development, especially in $BRAF^{V600E}$ mutation positive PTC.

In our GWAS, the most robust association was located near *NRG1*. Therefore, we could suppose that the *NRG1* loci might be the susceptible loci for PTC in this population. For instance, Gudmundsson et al.[10] reported that a significant correlation was observed between genotypes on *NRG1* and the relative expression of *NRG1* in blood, and we hypothesized that the expression level of *NRG1* in thyroid tissue might be determined by the identified variants. As expected, we also found that the *cis*-eQTL data showed that the variants in the *NRG1* region were associated with the expression of *NRG1* in both normal and cancer thyroid tissues from our RNA-sequencing results. The same association was found when we analysed the data using the predicted expression results or the public database, and those variants were associated with increased *NRG1* expression in normal thyroid tissue and whole blood. Furthermore, the GSEA identified that one of the variants rs6996585 was associated with many pathways related to cellular growth or cancer and the *ERBB-MAPK* signalling pathway was the most significantly enriched with its related signals (Fig. 2f and Supplementary Table 8). Our clinical results also showed that the variants of *NRG1* are associated with lymph node metastasis in thyroid cancer, especially in $BRAF^{V600E}$-mutated PTC. A recent report supported this association, showing an association between a variant of *NRG1* (rs2439302) and lymph node metastasis in PTC[26]. Although we did not demonstrate the direct effects of the increased expression of *NRG1* on tumour aggressiveness, we could postulate a possibility that the increased expression of *NRG1* in the thyroid tissue, which is associated with identified variants, might influence the development or progression of thyroid cancer. That is, the *NRG1* region could be an important risk gene for the susceptibility or prognosis of thyroid cancer. Although it is

uncertain why the *NRG1* region demonstrated the most significant association in our study, one plausible explanation could be that it was caused by the difference in MAF of the *NRG1* variant between the Asian population and other populations (Supplementary Table 4).

A mutation in *NKX2-1*, known as thyroid transcription factor (*TTF) 1*, is a causative mutation for brain–lung–thyroid syndrome characterized by congenital hypothyroidism, respiratory distress syndrome and benign hereditary chorea[27]. In a previous GWAS for DTC, one common variant (rs944289), located near *MBIP/NKX2-1*, was associated with the mentioned diseases[8,10] and several studies validated the association[15,16]. Jendrzejewski *et al.*[28] reported that the expression of a non-coding RNA gene named papillary thyroid carcinoma susceptibility candidate 3 (*PTCSC3*) located near *MBIP/NKX2-1* was downregulated in thyroid tumour tissue and the risk allele (T) was associated with the profound suppression, which implies that *PTCSC3* could have some role as a tumour suppressor in DTC. We confirmed that the variant in rs944289 and another variant in rs34081947 were related with PTC and the expression of *NKX2-1* was increased in thyroid tissues containing those variants (Table 2).

*FOXE1* (Forkhead Box E1), also known as *TTF2*, plays a role in the development or differentiation of the thyroid[29,30]. Landa *et al.*[31] demonstrated that variants near *FOXE1* affected *FOXE1* transcription through the recruitment of *USF1/USF2* transcription factors. The *FOXE1* locus was reported as a major genetic determinant of risk in DTC and radiation-related PTC in several GWASs of European descent. Specifically, the relationship between common variations of *FOXE1* and DTC was validated in several studies[8,9,32], and rs965513 of *FOXE1* is the most well-replicated susceptibility locus in the East-Asian population (Supplementary Table 1). In recent research, some variants of the *FOXE1* locus were associated with clinical phenotypes of PTC, such as tumour stage, size, lymphocytic infiltration and extrathyroidal extension[26,33]. Our results showed that only one variant (rs72753537) of *FOXE1* was significantly associated with PTC after replication, but there exist possibilities of six other variants as risk loci of DTC development (Supplementary Table 10). We also did not find any association between the variants and the *FOXE1* expression levels or clinical phenotypes despite the fact that the variants' association with *FOXE1* expression level was observed in the public database. *FOXE1* is known as the most susceptible gene of DTC. However, the effect of *FOXE1* variants on thyroid cancer risk seemed to be less significant than that of *NRG1* in this Asian population. Compared with the 1,000 genome database, there are significant ethnic differences in the allele frequencies for the variants of *FOXE1* between the European and Asian populations (MAF; 0.14–0.34 versus 0.08–0.13). This suggests that Asian populations with this polymorphism are relatively small and are thus less susceptible to thyroid cancer than European populations.

Given that the association between rs966423 on the *DIRC3* region and thyroid cancer was first reported in a European GWAS[10] and was replicated in limited ethnic groups, it is impossible to ignore the possibility of the presence of population heterogeneity[12]. Wang *et al.*[15] showed that the variant rs966423 on *DIRC3* was associated with an increased PTC risk, which was confirmed by our present study ((additive model of logistic regression) $P = 0.0067$, Supplementary Table 11). In a recent report, rs966423 was associated with increased mortality in DTC[34]. In this study, two SNPs were associated with DTC that were different SNPs from the previous report. Notably, rs1549438 was associated with both the FTC and the expression of *TNS1*.

As described above, the previous reported region of *NKX2-1*, *FOXE1* and *DIRC3* showed either a relatively less association than a European study, or the association was observed in different SNPs. However, these regions were still found to be risk loci in Korean or other Asian populations as in European populations. As for the regions in *IMMP2L*, *DHX35*, *ARSB* and *WDR11-AS1*, no association was observed (Fig. 4 and Supplementary Table 11).

Here we identified six novel susceptible loci near *VAV3*, *PCNXL2*, *INSR*, *MRSB3*, *FHIT* and *SEPT11* that were not found in previous European studies.

The *VAV3* gene encodes a guanine nucleotide exchange factor 3 for the Rho family of GTPases, which activates pathways involving actin cytoskeletal rearrangements and transcriptional alterations[35]. Casual variants on *VAV3* were associated with hypothyroidism in both a European and our previous Korean GWAS[36,37]. *VAV3* expression in thyroid cancer cells is also known to be RET/PTC and RAS mutation-specific because *VAV3* is involved in *PI3K* signalling and the subsequent *AKT* activation[38]. In this study, we first demonstrated that the association between the variation at *VAV3* and DTC risk or *VAV3* expression.

*MSRB3* encodes a zinc-containing methionine sulfoxide reductase B3 and its mutations are associated with human deafness[39]. Methionine sulfoxide reductase is suggested to utilize catalytic selenocysteine[40]. Selenocysteine is an essential component of deiodinases enzymes, which convert thyroxine (T4) into triiodothyronine (T3) and is also associated with thyroid autoimmunity[41]. Thus, variants in *MSRB3* could be related with the pathogenesis of thyroid cancer. The fusion of *MSRB3* was found in the primary and metastasis tumour spindle cells[42]. Although we could not find any association between the variant of *MSRB3* and *MSRB3* expression or clinical phenotypes, there was an association with FTC as well as with PTC.

Another newly identified susceptibility locus was located near *INSR*. In a previous meta-analysis of GWASs for thyroid-related traits, common variants in *INSR* were reported to be associated with the levels of thyroid-stimulating hormone[43]. Insulin is reported to be upregulated in various cancers[44,45] and in *in vitro* studies the expression of *INSR* was also elevated in malignant thyrocytes[46]. In this study, we found a risk variant of PTC in the *INSR* gene, although it was not so strong. However, the presence of a risk allele was significantly associated with lymph node metastasis, suggesting a possible pathogenic role in thyroid cancer.

The other 3 new genes are *FHIT* (Fragile Histidine Triad), *PCNXL2* and *SEPT11*. *FHIT* encodes a diadenosine 5′,5‴-P1, P3-triphosphate hydrolase and is known as a tumour suppressor gene in a variety of common human cancers[47,48]. In thyroid cancer, the inactivation of *FHIT* was suggested to be associated with the pathogenesis of thyroid neoplasm[49] and the homozygous deletion and promoter methylation of *FHIT* was reported to be associated with DTC[50,51]. This is concordant with our result, although the association between a variant in *FHIT* and PTC risk was not so strong, and no association with the gene expression or clinical phenotypes was observed.

*SEPT11* encodes septin-11, which is involved in a filament-forming cytoskeletal GTPase and may play a potential role in cytokinesis[52]. *SEPT11* is also known as a gene of the fusion partners of MLL in chronic neutrophilic leukaemia[53]. Regarding *PCNXL2*, there was one report suggesting that it might play a role in the tumourigenesis of colorectal carcinomas with a high microsatellite instability[54]. The biological functions of both *PCNXL2* and *SEPT11* on thyroid tissue were not defined. However, the expression of *PCNXL2* was increased in thyroid tissues with a candidate variant and there was an association between a candidate variant of *SEPT11* and extrathyroidal extension, suggesting their possible role in the development of thyroid cancer.

Very recently, Gudmundsson *et al.*[55] reported a GWAS yielding five novel risk loci for thyroid cancer. Among the five loci, one locus (1q42.2, *PCNXL2*) was included among the significant loci in our results (Table 1). However, the intron variant (rs12129938) of *PCNXL2* that was reported as most significant by Gudmundsson *et al.*[55] showed only a marginal association ($P = 0.002$) in our results. Instead, the other intronic variant (rs4649295) showed the most significant association ((additive model of logistic regression) Joint $P = 6.0 \times 10^{-8}$). The 10q24.33 locus (near *OBFC1*) also showed a suggestive association ((additive model of logistic regression) $P = 8.72 \times 10^{-6}$) in our results. The other novel risk loci (3q26.2, 5q22.1 and 15q22.33) and one replicating locus (5p15.33) showed no associated signal ($P < 0.001$) in our results. A comparison of reported SNPs and our results is summarized in Supplementary Table 12.

The limitation of this study is that we used two different genotyping platforms in the discovery stage because the genotyping of the control sample was performed in the past. As a consequence, potential SNPs were excluded from the study to remove a batch effect and information might have been lost in the process. Another limitation was the imbalance of gender between the case and control groups caused by a higher prevalence of thyroid cancer in the female population. To adjust for this imbalance, we repeated the test with a stratification for sex and, although the statistical significance was lowered, there was no difference in the candidate genes. Lastly, the control samples of the replication stage were from participants of a relatively higher age. However, it was an inevitable consequence of ensuring that there is no misclassification of the participants (to confirm that the control participants truly did not have cancer). Furthermore, these participants went through an ultrasonography examination to ensure they were 'super normal' controls and the result of the comparison between cancer and normal should thus be more reliable and be a unique value of this study.

The strength of this study is that this is the first study to be performed in an Asian population using GWAS in thyroid cancer and we identified some ethnic differences. When we compared the DTC SNPs reported in Europeans with our Stage 1 result, we found a similar effect size in the SNPs of *FOXE1* and *NRG1* between the two populations, but the effect of *NKX2-1* and *DIRC3* was lower in Koreans compared with Europeans and we suspect this may be due to a difference in the genetic susceptibility between different ethnic populations.

In addition, we discovered specific variants of DTC that have different effects on PTC compared to their effects on FTC. First, the variants of *NRG1* were more associated with PTC than DTC, showing the most robust effect. Second, a variant of *SLC24A6* showed a high risk effect (OR = 3.32) specific to FTC. Although both PTC and FTC are cancers of follicular cell origin, the mutational profiles are quite different between PTC and FTC, our results suggest that the risk assessment of thyroid cancer development should be tailored according to cancer type and personal genotypes.

Furthermore, we were the first to identify that the candidate SNPs influence the molecular and biological changes in the development of thyroid cancer by performing an eQTL analysis.

In conclusion, we conducted a GWAS for DTC in Koreans. We confirmed significant associations at previously reported loci of *NRG1*, *FOXE1*, *NKX2-1* and *DIRC3*, and that *NRG1* was the most significantly associated in this Asian population. We revealed novel susceptibility loci at *VAV3*, *PCNXL2*, *FHIT*, *SEPT11*, *MSRB3* and *INSR*. We also validated these results with a *cis*-eQTL analysis using RNA sequencing data from the tumour and normal thyroid tissues. We propose that these results can be implied for the diagnosis and treatment of thyroid cancer and provide more insight into genetic factors in the era of personalized medicine in cancer.

## Methods

**Study participants for the Stage 1 genome scan.** The DNA samples of the thyroid cancer cases for the Stage 1 genome scan were collected at the Seoul National University Hospital. These cases comprised 470 DTC patients (410 PTC and 60 FTC) who underwent a thyroidectomy. The baseline characteristics of the study subjects are shown in Supplementary Table 2. The controls comprised 8,279 subjects and were recruited from the KoGES (Korean Genome and Epidemiology Study) and the Ansung or Ansan cohort, of which an initial investigation began in 2001 with 8,842 participants aged 40–69 years[56]. The controls were not evaluated on thyroid disease. All of the participants in this study were of Korean ancestry (Supplementary Fig. 1).

**Study participants for the Stage 2 follow-up.** For validation of the candidate associations, we used independent case–control groups. The cases comprised 615 subjects; 524 of the samples (515 PTC and 9 FTC) were from the National Cancer Center and 91 of the samples (72 PTC and 19 FTC) were from Seoul National University Hospital. Six hundred and five controls were taken from the National Cancer Center and Seoul National University Hospital, respectively (Supplementary Fig. 1). The DNA samples from the cases were collected at the time of thyroidectomy and those from the controls were collected when they underwent a health check-up. All of the controls showed a normal thyroid in the ultrasonography examination or had pathologically proven benign nodules. For the eQTL study, 78 thyroid cancer cases (60 PTC and 18 FTC) that had RNA-sequencing results of their tumour or normal tissues were enrolled from Seoul National University Hospital. The baseline characteristics of the cases with thyroid cancer are shown in Supplementary Table 2. All of the subjects in the replication study were residents of Korea and of Korean ancestry.

**Discovery SNP genotyping and imputation.** The DNA was extracted from the leukocytes of peripheral blood samples obtained from the study individuals. For the Stage 1 genome scan, the thyroid cancer case samples were genotyped using the Illumina HumanCore-24 BeadChip kit (Illumina, San Diego, USA) and the control samples were genotyped using the Affymetrix Genome-Wide Human SNP Array 5.0 (Affymetrix Inc., Santa Clara, USA)[57]. To minimize the possible genotyping errors, the SNPs were excluded by the criteria defined by Hardy–Weinberg equilibrium ($P < 1 \times 10^{-6}$), the call rate ($< 95\%$) and the MAF ($< 1\%$). After strand alignment with PLINK v1.07 and phasing with SHAPEIT2, imputation was carried out using IMPUTE2 software in both of the cases and controls. The 1,000 genome ASN Phase I reference panel (integrated variant set release in NCBI build 37, hg19) was used as a reference. For imputation quality control, only variants with an Info Score $< 0.9$ were excluded. After merging the data sets from the cases and controls, we excluded SNPs with a missing genotype rate $\geq 5\%$ (1,590,137 SNPs excluded) and SNPs whose genotype frequencies were out of range from Hardy–Weinberg equilibrium at $P < 1 \times 10^{-6}$ (290 SNPs excluded). As we used two different platforms for genotyping (Illumina for cases and Affymetrix for controls), there would be spurious associations due to the batch effect[58]. To detect false associations, we used the SNP data from another healthy cohort comprised of 2,000 subjects, which was genotyped using an Illumina HumanCore-24 BeadChip. We tested for the batch effects by analysing the association between the SNPs from the controls and healthy cohorts. We selected for the P-values using a conservative threshold of $1 \times 10^{-3}$ and a total of 31,279 SNPs were excluded. Finally, 3,593,389 markers were used for selecting the candidate SNPs.

**Replication SNP selection and genotyping.** In total, forty-one independent variants were selected for the replication test. Among the candidate SNPs with $P < 2.0 \times 10^{-5}$ from the genome-wide scan, we selected 27 representative SNPs in 25 candidate regions based on the clustering pattern and the linkage dis-equilibrium. We also included 14 SNPs located in previously reported risk loci for thyroid cancer or thyroid disease (Supplementary Table 3).

For the Stage 2 follow-up genotyping, the selected SNPs were genotyped using the Fluidigm SNP Type Assay platform (Fluidigm, San Francisco, USA). To maintain the genotyping quality, a genotyping call rate of $> 95\%$ and a Hardy–Weinberg equilibrium with a $P > 0.001$ were considered.

**RNA sequencing and eQTL analysis.** The details of the RNA sequencing methods used were previously reported[59]. In brief, 78 tumour tissues and 23 normal tissues from the case samples of the replication stage were sequenced using a HiSeq 2000 platform (Illumina). Then, we profiled the gene expression according to the Ensembl gene set with the count number of reads aligned to each gene using the HTSeq-count and normalized them via fragments per kilobase of exons per million fragments mapped. To investigate the *cis*-eQTL of the chosen SNPs to neighbouring genes ($\pm 500$ kb), the RNA-sequencing profiled data were assessed according to the additive model of the linear regression analysis. We also evaluated the effect of the associated genotypes on expression in various tissues using the public eQTL database (GTEx[60] and Whole blood eQTL[61]). We conducted the

imputation of gene expression in 470 DTC samples using the PrediXcan package (https://github.com/hakyimlab/PrediXcan/tree/master/Software)[62]. The normal thyroid eQTL database (GTEx V6p, 278 thyroid samples) was used as a reference[60]. In patients with PTC from the discovery stage, several aggressive features including the presence of the $BRAF^{V600E}$ mutation, lymph node metastasis and extrathyroid extension were analysed according to the genotypes of GWAS-identified variants.

**Statistical analysis.** We conducted the case–control association analysis with the genome-wide SNPs via an additive model using PLINK software, version 1.9 (http://pngu.mgh.harvard.edu/~purcell/plink/)[63]. A logistic regression analysis was used to test the association for the series of GWAS, replication and joint analysis with unadjusted models, as well as with the adjustment for age and sex. To eliminate relatedness between each pair of subjects in the Stage 1 genome scan, the kinship identical-by-descent coefficient (Z0 > 0.8) was considered.

$Q$–$Q$ plots were used to assess the adequacy of the case-control matching. We also calculated the genomic inflation factor ($\lambda$) from a GWA analysis to compare the genome-wide distribution of the test statistics with the expected null distribution. The regional plots were created using LocusZoom (http://locuszoom.sph.umich.edu/locuszoom/).

The GSEA was conducted using GSEA software, version 2.2.1 (Broad Institute, www.broad.mit.edu/gsea/msigdb/index.jsp) with the BioCarta, KEGG and Reactome (1,077 gene sets) of Molecular Signatures Database (MSigDB version 5.1, http://www.broadinstitute.org/gsea/msigdb/)[64]. The IBM SPSS Statistics for Windows, version 23 (IBM Corp., Armonk, USA) was used for the statistical analyses.

**Ethics statement.** This study was approved by the Institutional Review Board of the Seoul National University Hospital (IRB Number H-1102-012-349 and H-1108-041-372) and the National Cancer Center (IRB Number NCC2015-0027), and written informed consents were obtained from all of the participants. All of the clinical investigations were conducted according to the principles expressed in the Declaration of Helsinki.

**Data availability.** The KoGES (Korean Genome and Epidemiology Study) genotype data that support the findings of discovery stage are available upon request under data sharing policy of National Research Institute of Health, Korea (http://www.nih.go.kr/). The RNA-sequencing data that support the findings of validation are available in EBI European Nucleotide Archive database with accession number PRJEB11591. Other data that support our findings are available from the corresponding author upon reasonable request.

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

## Acknowledgements

This work was supported by the National Research Foundation of Korea (NRF) grant funded by the Korea government (MSIP) (Grant Number 2012R1A5A2A44671346). This research was supported by a grant of the Korea Health Technology R&D Project through the Korea Health Industry Development Institute (KHIDI), funded by the Ministry of Health & Welfare, Republic of Korea (Grant Number HI13C2148). This study was supported by grant 04-2009-0780 from the SNUH Research Fund. This work was supported by a grant from Research Grant Numbers 1410650-2 and 1410640-3 to E.K.L. from the National Cancer Center. This work was supported by a grant from Research Grants Number NCC1510040 to J.K. from the National Cancer Center.

## Author contributions

J.-I.K., Y.J.P. and E.K.L. planned and managed this study. Y.J.P., S.W.C., D.J.P., S.-J.K., K.E.L., J.Y.L., N.H.C., J.S.R., Y.-S.J., T.H.K. and J.K. carried out data and sample collection. H.-Y.S., Y.H., S.-K.Y., E.K.L., J.S., S.H.K. and J.-S.S. carried out genotyping experiments, imputation, RNA sequencing and analysed the data. S.-W.I., S.D.Y., S.-J.K. and M.S.P. assisted and analysed the data, and wrote the manuscript. J.-I.K., Y.J.P., E.K.L., H.-Y.S. and Y.H. contributed to writing and editing the manuscript. All authors approved the final manuscript.

## Additional information

**Competing interests:** The authors declare no competing financial interests.

