## [Peer Review File · Nature Communications]

Reviewers' comments:

Reviewer #1 (Remarks to the Author):

Following previous genome-wide association studies the authors conducted a GWAS and a replication study using a total of 1,085 DTC cases and 8,884 controls of Koreans and validated their results with an expression quantitative trait loci (eQTL) analysis and clinical phenotypes. This is the first study of this type carried out among Asians.

The study is carried out in an excellent way and it is well written. The results are consistent with previous studies and actually they add more knowledge on the risk factors for thyroid carcinoma. The statistical power and the design of the study is appropriate.

In order to enrich the manuscript I would suggest to include more data of gene expression and analyses by consulting TCGA public database

Minor comments:

line 68 (Poles?)

line 163 there was also showed

Reviewer #2 (Remarks to the Author):

Son and colleagues describe a carefully performed and replicated/validated GWA and eQTL analysis in Korean DTC patients to identify several associated risk loci, which differ depending on ethnicity.

1. In this type of GWAS, finding risk loci is exactly that and I wouldn't describe them as susceptibility genes or even predisposing genes/alleles. The latter are too strong.
2. Now, if the authors wish to use a stronger word than risk loci, then they need to show segregation of their risk alleles in DTC families. That would be interesting even if one could show one or two very large families.
3. It would be interesting if these authors would take their GWAS hits and impute the differential gene expression using, eg, PrediXcan (or SLINGER or something similar) and see if the RNA data actually confirms the imputed transcriptomic changes, and whether in the right direction.
4. PTCSC2 and SRRM2 (He H et al. JCEM 2015; Tomsic J et al Sci Rep 2015) appear to be quite universal. Do the authors see associations therein?
5. Similarly, TPOAb loci (Medici M et al. PLoS Genet 2014) should be a fundamental mechanism – do the authors see associations in these regions? If not, then why not?

Response to Decision Letter

Reviewers' comments:

Reviewer #1:

Following previous genome-wide association studies the authors conducted a GWAS and a replication study using a total of 1,085 DTC cases and 8,884 controls of Koreans and validated their results with an expression quantitative trait loci (eQTL) analysis and clinical phenotypes.

This is the first study of this type carried out among Asians.

The study is carried out in an excellent way and it is well written. The results are consistent with previous studies and actually they add more knowledge on the risk factors for thyroid carcinoma. The statistical power and the design of the study is appropriate.

Major comments

Comment 1: In order to enrich the manuscript I would suggest to include more data of gene expression and analyses by consulting TCGA public database

Response:

We appreciate the reviewer for this thoughtful comment. It should be very helpful for validation of our findings to analyze TCGA public data together. As the reviewer commented, TCGA have somatic alterations data from 507 thyroid cancer and RNAseq data from 502 thyroid cancer.

First, we compared RNA expression patterns of reported genes between TCGA data and this study. We found entire pattern was similar as following figures.

TCGA data

This study

However, our data on eQTL could not be validated by TCGA data, because they only provides exonic genotype data and SNPs of our result were mostly located in intronic or intergenic region (Supplementary Table 3).

Instead, we could validate our findings to predicted expression result using PrediXcan package following the other reviewer's recommendation. We revised as the **comment 3 of reviewer #2**. We appreciate the reviewer's helpful and thoughtful comments.

Minor comments

Comment 1: line 68 (Poles?).

Response: We thank for the reviewer's pointing out. We changed the word "Poles" to "Polish".

Comment 2: line 163 'there was also showed'

Response: We have rephrased sentence as follows.

A similar expression pattern was shown in the normal thyroid tissue, although it was not statistically significant (Page 7)

Reviewer #2: Comments

Son and colleagues describe a carefully performed and replicated/validated GWA and eQTL analysis in Korean DTC patients to identify several associated risk loci, which differ depending on ethnicity.

Major comments

Comment 1: In this type of GWAS, finding risk loci is exactly that and I wouldn't describe them as susceptibility genes or even predisposing genes/alleles. The latter are too strong.

Response: Thank you for the reviewer's insightful comment. We agreed with the opinion that the "risk loci" fits for our findings rather than "susceptibility genes" or "predisposing one" and revised as the response of **comment 2**.

Comment 2: Now, if the authors wish to use a stronger word than risk loci, then they need to show segregation of their risk alleles in DTC families. That would be interesting even if one could show one or two very large families.

Response:

We thank for the reviewer's important comment. We agree with the reviewer's opinion on the genetic terms. The expression "susceptibility genes or predisposing genes" would be more appropriate to be used in family studies than GWAS as the reviewer pointed out. It is true that the segregation of their risk alleles in DTC families will further strengthen our findings. However, this GWAS was conducted only in population-based cohort that does not include family data. Since there is a possibility that the loci identified in the GWAS are to play a functional role in development of thyroid cancer, further investigations whether genomic alteration of the loci occurs in DTC families through NGS would be very interesting and informative. We changed the expression of "susceptible genes/variants/SNPs" throughout our manuscript as the reviewer recommended.

Novel candidate variants in the VAV3, PCNXL2, INSR, MRSB3, FHIT or SEPT11 loci (Page 9)

However, these regions were still found to be risk loci in Korean or other Asian populations (Page 13)

... there was no difference in the candidate genes. (Page 15)

Furthermore, we were the first to identify that the candidate SNPs influence the molecular and biological changes (Page 16)

Comment 3: It would be interesting if these authors would take their GWAS hits and impute the differential gene expression using, eg, PrediXcan (or SLINGER or something similar) and see if the RNA data actually confirms the imputed transcriptomic changes, and whether in the right direction.

Response:

We appreciate the reviewer's important comment, which improve our manuscript by validating the eQTL results. We could validate our e-QTL results using PrediXcan package, and added or changed the results to our manuscript as follows.

Result

Additionally, we evaluated the cis-eQTL of the normal thyroid tissues (Supplementary Table 5) and other various tissues (Supplementary Table 6) in the public eQTL database and imputed expression (Table 2) of 470 DTC case samples using PrediXcan, found similar results with our RNAseq eQTL data. (Page 6-7)

However, the predicted expression result of discovery case samples that used normal thyroid reference data, showed a highly significant association ($P = 2.99 \times 10^{-244}$, Fig. 2c, Table 2). (Page 7)

However, the predicted expression result showed a cis-eQTL with C9orf156, CORO2A, XPA and TSTD2 genes (Table 2) and the public expression data showed a cis-eQTL with C9orf156 in various tissues. (Page 9)

The predicted expression result showed a highly significant association ($P < 1.00 \times 10^{-300}$, Table 2) and the public expression data also showed a significant association ($P = 3.33 \times 10^{-27}$) in thyroid tissue. (Page 9)

However, we could not find any cis-regulation of nearby gene expression in the FHIT, SEPT11, MSRB3 and INSR regions with our expression data, but the predicted expression result showed a cis-eQTL for INSR ($P = 4.91 \times 10^{-41}$, Table 2) and the public expression data showed... (Page 9)

Discussion

It was the same when we analysed the data using the predicted expression result or public database, ... (Page 11)

Methods

We conducted the imputation of gene expression in 470 DTC samples using PrediXcan package (<https://github.com/hakyimlab/PrediXcan/tree/master/Software>)⁶². The

normal thyroid eQTL database (GTEx V6p, 278 thyroid samples) was used as a reference⁶⁰. (Page 19)

Reference

62 *Gamazon, E.R. et al. A gene-based association method for mapping traits using reference transcriptome data. Nat Genet 47, 1091-8 (2015).*

Table 2, the eQTL result using public data (GTEx) was moved to the Supplementary Table 5 and predicted expression result using PrediXcan was included.

Figs 2 and 3, eQTL figures of GTEx were moved to Supplementary Figure3 and predicted expression box plots were included.

Comment 4: PTCSC2 and SRRM2 (He H et al. JCEM 2015; Tomsic J et al Sci Rep 2015) appear to be quite universal. Do the authors see associations therein?

Response:

We appreciate the kind comments. The *FOXE1* locus is a major genetic determinant for thyroid cancer in European descendants. Papillary thyroid cancer susceptibility candidate 2 (*PTCSC2*) is long intergenic noncoding RNA gene, which located on the opposite strand of *FOXE1* in the human genome. Exon 1 and intron 1 of transcript isoform C of *PTCSC2* overlap with the promoter region of *FOXE1*. Recent study showed myosin-9 (*MYH9*) binds to *PTCSC2* and regulates the bidirectional promoter shared by *PTCSC2* and *FOXE1* (Wang et al. PNAS 2017). We selected 7 SNP from *FOXE1/PTCSC2* locus and conducted replication series (Supplementary Table 10). However, most of SNP were not replicated. And *cis*-eQTL analysis could not find significant association with *PTCSC2* as follows.

PTCSC2 eQTL			Normal thyroid tissue			Tumor thyroid tissue		
CHR	SNP	BP	N	BETA	P-value	N	BETA	P-value
9	rs1588635	100537802	23	0.1366	0.9796	78	-0.5954	0.8881
9	rs965513	100556109	23	0.1366	0.9796	78	-0.5954	0.8881
9	rs1867277	100615914	22	-0.0576	0.9888	75	1.458	0.7157
9	rs72753537	100660746	23	5.33	0.3112	78	-7.336	0.2754
9	rs7028661	100538470	23	0.1366	0.9796	78	-0.5954	0.8881
9	rs10122541	100628268	23	5.33	0.3112	78	-5.356	0.3938
9	rs7037324	100658318	23	5.33	0.3112	78	-5.356	0.3938

A germline mutation c.1037C > T (Ser346Phe or S346F; rs149019598) in serine/arginine repetitive matrix 2 (*SRRM2*), which was identified in familial PTC, was newly suggested as genetic variants for PTC. But the variant on rs149019598 was found in only 7/1,170 sporadic PTC cases and in 0/1,404 controls. 1000genome also showed that global minor allele frequency of rs149019598 is extremely rare (0.00019). In this GWAS, we excluded SNPs with MAF < 1%, thus we could not identify rs149019598 in discovery series (Methods - Discovery SNP genotyping and imputation). SNPs near rs149019598 (nearby 500kb) did not show meaningful association with PTC ($P > 0.0001$).

We did not include these data in this revision, but we are willing to add if the reviewer recommend.

Comment 5: Similarly, TPOAb loci (Medici M et al. PLoS Genet 2014) should be a fundamental mechanism – do the authors see associations in these regions? If not, then why not?

Response:

We appreciate the reviewer's important comment. We also looked at 5 loci (*TPO*, *ATXN2*, *BACH2*, *MAGI3*, *KALRN*), which were reported in the above paper (Medici M et al. PLoS Genet 2014) (we analyzed TPO instead the TPOAb). However, there was no association with $P < 0.001$ in *TPO*, *ATXN2*, *BACH2*, *MAGI3* loci and most strong association in *KALRN* locus have P -value 0.00031 in our data.

On the other hands, TSH level has been known to be closely related with risk of thyroid cancer (Haymart MR et al. J Clin Endocrinol Metab. 2008). Common genetic loci, which have association both with TSH level and thyroid cancer include *NXK2-1*, *NRG1*, *FOXE1* and *VAV3*. Thus, we think that TSH level, risk of thyroid cancer, and several genetic factors were closely related with each other.

There also has been proposed an association between Hashimoto's thyroiditis and thyroid cancer. As the reviewer commented, it would be very interesting if we could find shared susceptible loci between them. Regarding TPOAb positivity, there exists one GWAS result demonstrating 4 susceptible loci; rs301799 near *RERE* and 3 markers in the *HLA* region (rs3094228, rs1894407, and rs9277555) (Schultheiss UT et al. J Clin Endocrinol Metab. 2015). However, we could not find any significant association of those 4 genetic locus in our thyroid cancer subjects as follows. All these result was not included in current revision, but we are willing to add if the reviewer recommend to include.

Chr	SNP	Position	Nearby gene	P -value of TPOAb concentration / Positivity	Our result		
					P -value	Nearby top SNP with P < 0.001	P -value of nearby top SNP
1	rs301799	8,489,302	RERE	2.5E-05 / 1.4E-08	-	rs2781085	0.00018
6	rs3094228	31,429,927	HCP5, MICA	2.7E-07 / 1.3E-08	-	-	
6	rs1894407	32,787,036	HLA-DOB, TAP2	4.0E-08 / 1.4E-08	0.7851	rs2581	0.0005
6	rs9277555	33,055,605	HLA-DPBI	3.9E-12 / 3.0E-10	-	rs2581	0.0005

We really appreciate the reviewer's kind and important comment, again.

REVIEWERS' COMMENTS:

Reviewer #1 (Remarks to the Author):

I think the authors did their best to make the results stronger than before and the results are genuine.

Risk loci described in this manuscript could be important and worth of publication.

I am positive for the manuscript to be published.

Reviewer #2 (Remarks to the Author):

The authors have been responsive to original review. However, the paper could be made more readable if the author could invite a colleague fluent in English and the topic to edit for proper English expression, usage and grammar.

Response to Decision Letter

Reviewers' comments:

Reviewer #1:

I think the authors did their best to make the results stronger than before and the results are genuine. Risk loci described in this manuscript could be important and worth of publication.

I am positive for the manuscript to be published.

Response: We appreciate the reviewer for your kind review.

Reviewer #2:

The authors have been responsive to original review. However, the paper could be made more readable if the author could invite a colleague fluent in English and the topic to edit for proper English expression, usage and grammar.

Response: Thank you for the reviewer's comment. We revised our manuscript for this submission with the help of English-Language Editing Service of American Journal Experts (AJE).